# Prognostic Thresholds of Mitotic Count and Ki-67 Labeling Index for Recurrence and Survival in Lung Atypical Carcinoids

**DOI:** 10.3390/cancers16030502

**Published:** 2024-01-24

**Authors:** Patrick Soldath, Daniel Bianchi, Beatrice Manfredini, Andreas Kjaer, Seppo W. Langer, Ulrich Knigge, Franca Melfi, Pier Luigi Filosso, René Horsleben Petersen

**Affiliations:** 1European Neuroendocrine Tumor Society Center of Excellence, Copenhagen University Hospital-Rigshospitalet, 2100 Copenhagen, Denmark; andreas.kjaer@regionh.dk (A.K.); seppo.langer@regionh.dk (S.W.L.); ulrich.peter.knigge@regionh.dk (U.K.); rene.horsleben.petersen@regionh.dk (R.H.P.); 2Department of Clinical Medicine, University of Copenhagen, 2200 Copenhagen, Denmark; 3Department of Cardiothoracic Surgery, Copenhagen University Hospital-Rigshospitalet, 2100 Copenhagen, Denmark; 4Division of Thoracic Surgery, Department of Medical and Surgical Sciences, University of Modena and Reggio Emilia, 41121 Modena, Italy; danb9106@gmail.com (D.B.); pierluigi.filosso@unimore.it (P.L.F.); 5Minimally Invasive and Robotic Thoracic Surgery, Surgical, Medical, Molecular, and Critical Care Pathology Department, University of Pisa, 56126 Pisa, Italy; beatrice.manfredini91@gmail.com (B.M.); franca.melfi@unipi.it (F.M.); 6Department of Clinical Physiology and Nuclear Medicine & Cluster for Molecular Imaging, Copenhagen University Hospital-Rigshospitalet, 2100 Copenhagen, Denmark; 7Department of Biomedical Sciences, University of Copenhagen, 2200 Copenhagen, Denmark; 8Department of Oncology, Copenhagen University Hospital-Rigshospitalet, 2100 Copenhagen, Denmark; 9Department of Surgical Gastroenterology, Copenhagen University Hospital-Rigshospitalet, 2100 Copenhagen, Denmark; 10Department of Endocrinology, Copenhagen University Hospital-Rigshospitalet, 2100 Copenhagen, Denmark

**Keywords:** lung neuroendocrine neoplasms, atypical carcinoid, supra-carcinoid, prognostic biomarkers

## Abstract

**Simple Summary:**

Atypical carcinoid is a rare type of lung cancer, which exhibits a varying malignant potential. In this study, we aimed to identify the prognostic thresholds of the mitotic count and Ki-67 labeling index for recurrence and survival in atypical carcinoids. Our findings show that more patients recurred and died from this disease if their mitotic count exceeded three and four mitoses per 2 mm^2^, respectively, or if their Ki-67 labeling index exceeded 14% and 11%, respectively. These thresholds may serve as a valuable tool for clinicians and researchers in making treatment plans and predicting outcomes for patients with atypical carcinoids.

**Abstract:**

Atypical carcinoid (AC) is a rare neuroendocrine neoplasm of the lung, which exhibits a varying malignant potential. In this study, we aimed to identify the prognostic thresholds of the mitotic count and Ki-67 labeling index for recurrence and survival in AC. We retrospectively reviewed 78 patients who had been radically resected for AC and calculated said thresholds using time-dependent receiver operating characteristic curves and the Youden index. We then dichotomized the patients into groups of above or below these thresholds and estimated the cumulative incidences of the groups using the Aalen–Johansen estimator. We compared the groups using univariable and multivariable Fine–Gray subdistribution hazard models. Our findings show that more patients recurred and died from this disease if their mitotic count exceeded three and four mitoses per 2 mm^2^, respectively, or if their Ki-67 labeling index exceeded 14% and 11%, respectively. Both thresholds independently predicted survival (*p* < 0.001 and *p* = 0.015, respectively). These thresholds may serve as a valuable tool for clinicians and researchers in making treatment plans and predicting outcomes for patients with AC.

## 1. Introduction

Atypical carcinoid (AC) is a rare tumor belonging to the spectrum of lung neuroendocrine neoplasms. Unlike the other neoplasms of the spectrum, AC exhibits a varying malignant potential ranging from the indolent nature of typical carcinoid to the aggressive and metastasizing nature of large cell neuroendocrine carcinoma. This trait of AC has been explored over the past few years, which has led to the discovery of a subgroup of AC with carcinoid-like morphology but molecular and clinical features of large cell neuroendocrine carcinoma [1,2,3,4]. This new subgroup has been dubbed supra-carcinoid and is recognized in the newest edition of the World Health Organization (WHO) classification of thoracic tumors (5th edition, 2021) as AC with elevated mitotic count and/or Ki-67 labeling index (Ki-67 index) [5]. However, the diagnostic thresholds of these biomarkers remain undetermined. In this study, we aimed to identify the best prognostic thresholds of the mitotic count and Ki-67 index for recurrence and survival in AC to aid clinicians and researchers in establishing the diagnostic thresholds for segregating supra-carcinoids from AC.

## 2. Methods

### 2.1. Study Design

We reviewed three prospectively maintained databases of patients with AC who had been treated at three centers of thoracic surgery located in Copenhagen (Denmark), Modena (Italy), and Pisa (Italy) between July 2009 and January 2023. The study was approved by each center’s institutional review board, as well as the Danish and Italian health authorities. Informed patient consent was not required.

We included patients in the study if they had been radically resected for AC and not treated with neoadjuvant therapy. We extracted the patients’ demographic, clinical, and pathological characteristics from our institutional databases and their recurrence and survival status as well as cause of death from medical records and the Danish and Italian death indexes. All patients were diagnosed, treated, and followed up according to the guidelines of the European Society for Medical Oncology [6] while they were evaluated for surgery and operated as previously described in [7,8]. In short, the patients were preferably operated with lung-sparing, anatomical resections. All anatomical resections were followed by a systematic nodal dissection as outlined by the International Association for the Study of Lung Cancer [9], while wedge resections were followed a selective nodal sampling at the surgeon’s discretion. Tumors were classified as AC if they showed carcinoid-like morphology and more than 2 mitoses per 2 mm^2^ and/or necrosis as stated in the latest three editions of the WHO classification of thoracic tumors [5], which have been applicable during the study period. The patients were originally staged according to the 6th, 7th, or 8th edition of the American Joint Committee on Cancer staging manual [10]; thus, we restaged all patients according to the current 8th edition.

### 2.2. Biomarkers

We extracted three biomarkers from our institutional databases: the mitotic count, Ki-67 index, and necrosis. All biomarkers were assessed postoperatively on surgical specimens of the primary tumor by expert pulmonary pathologists. The surgical specimens were prepared and stained following the same protocol across the three centers [11]. Both the mitotic count and Ki-67 index were counted manually in hotspot regions. The mitotic count was recorded as the average number of mitoses per 2 mm^2^ of at least 5 distinct areas, while the Ki-67 index was recorded as the highest percentage of cells with positive nuclear labeling in a 20× field. Necrosis was classified as either none or focal.

### 2.3. Follow-Up

All patients were followed up with contrast-enhanced computed tomography (CT) scans of the chest and upper abdomen at three months, six months, and one year after surgery and then yearly for a minimum of ten years. Patients with central tumors were further followed up with bronchoscopy at six months, one year after surgery, and then yearly for five years. Patients who were suspected of recurrence via either CT scan or bronchoscopy were evaluated with somatostatin receptor (SSTR)-positron emission tomography (PET)/CT scan and, if technically feasible, a subsequent biopsy or resection of the lesion. Recurrence was diagnosed from a positive pathological specimen or a positive lesion on an SSTR-PET/CT scan. Locoregional recurrence was specified as lesions at the resection site or at the ipsilateral hilar or mediastinal lymph nodes. Distant recurrence was specified as lesions involving all other locations, including pleural and pericardial effusions.

### 2.4. Statistical Analysis

First, we assessed the relationship between the mitotic count and Ki-67 index using linear regression and the coefficient of determination (R^2^). We calculated the latter and its 95% confidence interval (95% CI) using bootstrap resampling with ten thousand resamples. Next, we analyzed each biomarker separately for recurrence and survival using competing risks models that counted either recurrence or death from AC as events and death from other causes as a competing risk. We censored the recurrence status at the time of the last follow-up scan and the survival status at the time of our retrospective review. We estimated the median follow-up times of recurrence and survival using the reverse Kaplan–Meier method, and we calculated the best prognostic thresholds of the mitotic count and Ki-67 index for recurrence and survival at five years after surgery using time-dependent receiver operating characteristic (ROC) curves and the Youden index. We then dichotomized the patients into groups of above or below these thresholds and estimated the cumulative incidences of the groups using the Aalen–Johansen estimator. We compared the groups using univariable Fine–Gray subdistribution hazard models, which we also used to evaluate whether necrosis, the pathological T-category, and the pathological N-category affected recurrence and survival. Finally, we evaluated whether any of these variables independently predicted recurrence and survival in multivariable models. As the mitotic count and Ki-67 index were correlated, we evaluated them in separate models. We considered *p*-values less than or equal to 0.05 statistically significant and analyzed all our data with R: A Language and Environment for Statistical Computing (version 4.3.2; R Core Team 2023; R Foundation for Statistical Computing, Vienna, Austria). The full reproducible code is available in the Appendix A.

## 3. Results

This study included 78 patients, of whom 26 had been treated in Copenhagen, 14 in Modena, and 38 in Pisa. The patients’ demographic, clinical, and pathological data are summarized in Table 1. Notably, all but 9 patients were anatomically resected. The mitotic count was available in 77 patients, while the Ki-67 index was available in 75 patients. The mitotic count ranged from 1 to 10 mitoses per 2 mm^2^ (median 3, interquartile range [IQR] 2–5), while the Ki-67 index ranged from 2% to 40% (median 10%, IQR: 8–18%). They were slightly correlated with an R^2^ of 0.31 (95% CI 0.13–0.52, *p* < 0.001). The R^2^ is visualized in Appendix A. 

The median follow-up times were 6.7 years for recurrence and 7.2 years for survival. No patients were lost to follow-up. During follow-up, 23 patients recurred, of whom 2 presented with locoregional recurrence (1 in lung and 1 in hilar nodes), 19 with distant recurrence (13 in liver, 3 in bones, 2 in brain, and 1 in pleura), and 2 with both locoregional and distant recurrences (both in mediastinal nodes, liver, and bones). Likewise, 20 patients died, of whom 14 died from AC and 6 from other causes. All but one of the patients who underwent recurrence had been anatomically resected.

The best prognostic thresholds of the mitotic count were three mitoses per 2 mm^2^ for recurrence and four for survival. The threshold for recurrence yielded a sensitivity of 78% and a specificity of 69%, while the threshold for survival yielded a sensitivity of 88% and a specificity of 83%. Similarly, the best prognostic thresholds of the Ki-67 index were 14% for recurrence and 11% for survival. The former yielded a sensitivity of 65% and specificity of 70%, while the latter yielded a sensitivity of 88% and a specificity of 58%. The ROC curves and their area under the curve (AUC) are shown in Figure 1.

More patients recurred and died from AC when their mitotic count exceeded three and four mitoses per 2 mm^2^, respectively, or when their Ki-67 index exceeded 14% and 11%, respectively. The cumulative incidences of recurrence and death from AC at five years from surgery were 49% and 43%, respectively, for patients who had a mitotic count above the thresholds, as well as 11% and 2%, respectively, for patients who had a mitotic count below the thresholds. The same incidences were 45% and 24%, respectively, for patients who had a Ki-67 index above the thresholds, as well as 19% and 4%, respectively, for patients who had a Ki-67 index below the thresholds. The cumulative incidence functions are shown in Figure 2 and Figure 3.

Both thresholds as well as focal necrosis and N2 disease predicted recurrence and survival in the univariable models, while T3–T4 disease only predicted recurrence. The univariable models for recurrence and survival are listed in Table 2. All variables remained statistically significant in the multivariable models except focal necrosis, which only remained as such for survival in the mitotic count model. In all cases, the effect sizes remained largely unchanged. The multivariable models for the mitotic count and Ki-67 index are listed in Table 3 and Table 4.

## 4. Discussion

Long before the unveiling of the supra-carcinoid [4], AC has been associated with a wide-ranging malignant potential. Our study is the first study to identify the prognostic thresholds of the mitotic count and Ki-67 index for recurrence and survival in AC. All identified thresholds classified the patients into groups of disparate prognoses and independently predicted both recurrence and survival. The thresholds varied slightly between recurrence and survival, most likely because the events of recurrence outnumbered the events of death from AC by more than one third in our relatively small cohort. The fewer events of death from AC and consequential greater uncertainty may also be reflected in the higher AUCs of the ROC curves for survival. The threshold of the mitotic count for survival was only one unit higher than the threshold for recurrence; however, this one-unit increase lowered the patients at risk for the groups above the thresholds by one third. This substantial shift in patients at risk occurred because the mitotic count spanned only ten successive values, and one of the thresholds was the median. On the other hand, the Ki-67 index comprised 20 distinct values across a range of 38 values; thus, the three-unit decrease from the threshold for recurrence to the threshold for survival merely shifted the patients at risk for the groups above the thresholds by a few.

To our knowledge, only one other study has investigated the thresholds of mitotic count and the Ki-67 index in AC [12]. That study included 153 patients with AC and found no statistically significant differences in the overall survival and disease-free survival between patients who had a mitotic count and Ki-67 index above or below the thresholds. However, these thresholds were not calculated by ROC curve analysis but instead chosen from the mean values of the mitotic count (five mitoses per 2 mm^2^ and 4.6 standard deviation [SD]) and Ki-67 index (10% and 12.2 SD). Furthermore, the Ki-67 index was only available in 69 out of the 153 patients. Nevertheless, these thresholds were quite close to our thresholds and thus the other study arguably might have been able to find differences in the recurrence and survival if the authors had used competing risks analyses instead of Kaplan–Meier analyses. The authors did not report the proportion of patients who died from AC and other causes. In our study, more than one quarter of the patients who died, died from other causes than AC. Assuming the same was true for the other study, these unrelated deaths might have prevented the study from finding statistically significant differences.

To this date, supra-carcinoid remains an unofficial entity that designate neuroendocrine neoplasms that show carcinoid-like morphology but also possess the molecular and clinical features of large cell neuroendocrine carcinoma. Thus, the diagnosis of supra-carcinoid currently demands molecular analysis. However, supra-carcinoid is often pragmatically referred to as AC with a mitotic count above 10 mitoses per 2 mm^2^ or a Ki-67 index value above 30% [13]. In our study, only four patients had a Ki-67 index value above 30%, while no patients had a mitotic count above 10 mitoses per 2 mm^2^. Although we only had four patients who fulfilled the unwritten criteria for supra-carcinoid, the patients in our groups with a mitotic count and Ki-67 index above the thresholds recurred and died from AC in rates that resemble those seen in large cell neuroendocrine carcinoma [14]. Even though we did not perform mutational analyses, our results seem to suggest that the unwritten criteria for segregating supra-carcinoid from AC may be overestimated and arguably should be lowered. However, another possibility is that a third subgroup may exist between AC and supra-carcinoid with a mitotic count and Ki-67 index value ranging from our thresholds to the unofficial thresholds of supra-carcinoid. To answer these questions, future studies are needed to investigate the molecular characteristics of AC of different mitotic counts and Ki-67 indexes. Although, our study is limited by a retrospective design, it is strengthened by a large uniform cohort of patients who had been treated at three European centers of thoracic surgery in a recent period using the very same approaches across all treatment aspects.

## 5. Conclusions

In conclusion, our identified thresholds of the mitotic count and Ki-67 index may serve as a valuable tool for clinicians and researchers in making treatment plans and predicting outcomes for patients with AC.

## Figures and Tables

**Figure 1 cancers-16-00502-f001:**
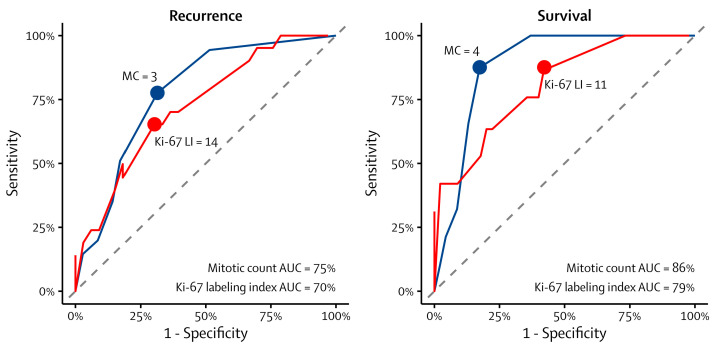
Receiver operating characteristic curves of the mitotic count and Ki-67 labeling index for recurrence (**left**) and survival (**right**). The blue and red curves represent the mitotic count and Ki-67 labeling index, respectively. The gray dashed line represents a random classifier. MC: mitotic count, Ki-67 LI: Ki-67 labeling index, and AUC: area under the curve.

**Figure 2 cancers-16-00502-f002:**
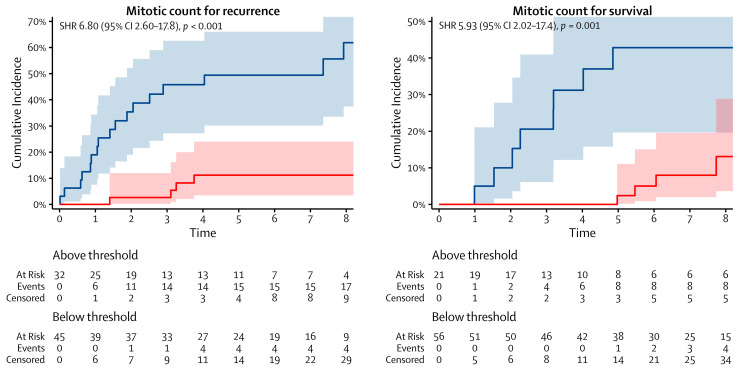
Cumulative incidence functions of the mitotic count for recurrence (**left**) and survival (**right**). The blue and red lines represent the cumulative incidence functions of the patients who were above and below the thresholds, respectively. SHR: subdistribution hazard ratio and CI: confidence interval.

**Figure 3 cancers-16-00502-f003:**
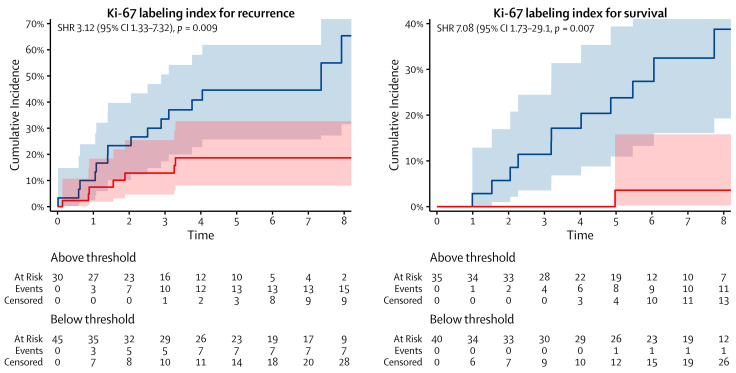
Cumulative incidence functions of the Ki-67 labeling index for recurrence (**left**) and survival (**right**). The blue and red lines represent the cumulative incidence functions of patients who were above and below the thresholds, respectively. SHR: subdistribution hazard ratio and CI: confidence interval.

**Table 1 cancers-16-00502-t001:** Patient characteristics. RATS: robotic-assisted thoracic surgery. VATS: video-assisted thoracic surgery. Values are n (%) and median (25–75%).

Characteristic	N = 78 ^1^
Sex	
Female	52 (67)
Male	26 (33)
Age	68 (62–72)
Smoking	
Current	17 (22)
Former	30 (38)
Never	31 (40)
Tumor location	
Central	32 (41)
Peripheral	46 (59)
Surgical resection	
Lobectomy	54 (69)
Bilobectomy	1 (1.3)
Sleeve lobectomy	6 (7.7)
Segmentectomy	7 (9.0)
Wedge	9 (12)
Bronchial resection	1 (1.3)
Surgical approach	
RATS	14 (18)
VATS	23 (29)
Thoracotomy	41 (53)
Pathological T-category	
T1a	10 (13)
T1b	25 (32)
T1c	11 (14)
T2a	12 (15)
T2b	9 (12)
T3	9 (12)
T4	2 (2.6)
Pathological N-category	
N0	55 (72)
N1	11 (14)
N2	10 (13)
Unknown	2
Pathological stage	
IA	36 (47)
IB	7 (9.2)
IIA	8 (11)
IIB	12 (16)
IIIA	11 (14)
IIIB	2 (2.6)
Unknown	2

^1^ n (%) and median (25–75%).

**Table 2 cancers-16-00502-t002:** Univariable Fine–Gray subdistribution hazard models for recurrence and survival. SHR: subdistribution hazard ratio and CI: confidence interval.

	Recurrence				Survival			
Characteristic	N	SHR ^1^	95% CI ^1^	*p*-Value	N	SHR ^1^	95% CI ^1^	*p*-Value
Mitotic count								
Below threshold	45	—	—		56	—	—	
Above threshold	32	6.80	2.60–17.8	<0.001	21	5.93	2.02–17.4	0.001
Ki-67 labeling index								
Below threshold	45	—	—		45	—	—	
Above threshold	30	3.12	1.33–7.32	0.009	30	6.08	1.77–20.9	0.004
Necrosis								
None	44	—	—		44	—	—	
Focal	33	3.69	1.44–9.46	0.007	33	4.63	1.25–17.1	0.022
T-category								
T1	46	—	—		46	—	—	
T2	21	1.48	0.56–3.91	0.43	21	0.99	0.26–3.78	0.99
T3 and T4	11	4.14	1.45–11.9	0.008	11	1.80	0.53–6.12	0.35
N-category								
N0	55	—	—		55	—	—	
N1	11	0.80	0.20–3.15	0.74	11	1.41	0.28–7.15	0.68
N2	10	6.48	2.38–17.7	<0.001	10	8.70	2.66–28.5	<0.001

^1^ SHR = subdistribution hazard ratio and CI = confidence interval.

**Table 3 cancers-16-00502-t003:** Multivariable Fine–Gray subdistribution hazard models of the mitotic count for recurrence and survival. SHR: subdistribution hazard ratio and CI: confidence interval.

	Recurrence				Survival			
Characteristic	N	SHR ^1^	95% CI ^1^	*p*-Value	N	SHR ^1^	95% CI ^1^	*p*-Value
Mitotic count								
Below threshold	44	—	—		55	—	—	
Above threshold	31	9.05	3.57–22.9	<0.001	20	6.65	2.36–18.7	<0.001
Necrosis								
None	42	—	—		42	—	—	
Focal	33	1.83	0.71–4.71	0.21	33	4.42	1.00–19.6	0.050
T-category								
T1	44	—	—					
T2	20	2.15	0.76–6.04	0.15				
T3 and T4	11	4.64	1.72–12.6	0.003				
N-category								
N0	54	—	—		54	—	—	
N1	11	0.29	0.06–1.46	0.13	11	1.14	0.23–5.71	0.88
N2	10	8.69	2.75–27.5	<0.001	10	7.74	2.84–21.0	<0.001

^1^ SHR = subdistribution hazard ratio and CI = confidence interval.

**Table 4 cancers-16-00502-t004:** Multivariable Fine–Gray subdistribution hazard models of the Ki-67 labeling index for recurrence and survival. SHR: subdistribution hazard ratio and CI: confidence interval.

	Recurrence				Survival			
Characteristic	N	SHR ^1^	95% CI ^1^	*p*-Value	N	SHR ^1^	95% CI ^1^	*p*-Value
Ki-67 labeling index								
Below threshold	43	—	—		38	—	—	
Above threshold	29	5.30	2.13–13.2	<0.001	34	5.65	1.41–22.7	0.015
Necrosis								
None	42	—	—		42	—	—	
Focal	30	1.78	0.75–4.21	0.19	30	4.10	0.82–20.4	0.085
T-category								
T1	43	—	—					
T2	19	0.59	0.14–2.49	0.47				
T3 and T4	10	6.34	2.51–16.0	<0.001				
N-category								
N0	53	—	—		53	—	—	
N1	9	1.77	0.31–10.1	0.52	9	1.50	0.30–7.57	0.63
N2	10	8.87	2.83–27.8	<0.001	10	8.25	2.80–24.3	<0.001

^1^ SHR = subdistribution hazard ratio and CI = confidence interval.

## Data Availability

The datasets generated during and/or analyzed during the current study are available from the corresponding authors on reasonable request.

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
