# Peer review of "Prognostic Thresholds of Mitotic Count and Ki-67 Labeling Index for Recurrence and Survival in Lung Atypical Carcinoids"

_cancers, 2024, doi:10.3390/cancers16030502_

Round 1

Reviewer 1 Report

Comments and Suggestions for Authors

1.  My main concerns are with the approach to data analyses.  

Supplemental graph -  Use of Spearman's Rho is less satisfactory than r-squared (coefficient of determination).  Only 1/3 of the variance is potentially associated with the variables being compared.   The data are considerably weighted by lower values of mitotic count.  Thus I would dispute the claim that the relationship strength is moderate (manuscript lines 134-136).

Figure 1 - Please provide confidence intervals around the plots for recurrence and survival in each of the two panels. (L 159 - 161) to support the claims concerning recurrence and death.

Tables 2-4 - Please provide "n" within the Tables for each of the hazard ratio determinations.

You may want to revisit your Results and Discussion sections for possible revision after you address the issues raised above concerning your approach to data analyses.

Author Response

Thank you for reviewing our manuscript.

  1. We have changed the correlation analysis to the R-squared method. We have updated the statistical analysis and results sections to reflect the new method. We have also updated the supplementary plot so it now annotates the R-squared with 95% confidence interval and p-value.
  2. We have updated the plots so they now show the 95% confidence intervals. 
  3. We have updated tables 2-4 so they now show the n.

Reviewer 2 Report

Comments and Suggestions for Authors

In their article authors made a thorough exploration of prognostic significance of mitotic rate and Ki67 expression in atypical carcinoid. The idea is based on a previously shown phenomena of the heterogeneity of this neoplasm. While the article does not shed light on biological mechanisms underlying the differences in clinical behaviour, authors conducted an important step in identification of a supra-carcinoid as an isolated condition with an alternative prognosis.

The article is well written. Includes significant number of rare condition and provides and outstanding follow up with diferentiating oncological and non-oncological outcome. All the results are supported with sufficient and thorough statistical analysis.

Author Response

Thank you very much for taking the time to review our manuscript.

Reviewer 3 Report

Comments and Suggestions for Authors

The manuscript is interesting and the presented data, even predictable, are important in the field and may be good source for further citations.

Specific comments

Introduction

In the introduction, the authors should describe primary and secondary goals 

Materials and methods

The research group is small. The research would be more valuable with a larger research group.

Discussion

There are too few comparisons of your results with the results of other authors.

I did not detect plagiarism. I do not have Ethic concern. I did not detect image manipulation.

Personally I would consider the paper for publication after the authors have addressed the major questions

Author Response

Thank you very much for reviewing our manuscript. Please see our responds to your comments below:

  1. In the end of the introduction we describe our aim.
  2. We agree that our cohort is small. Atypical carcinoid is a very rare cancer and therefore it is challenging to investigate large cohorts. Therefore, we decided to gather the data from our three centers and conduct a multicenter study. Of course it would be better if we included from even more centers, however, we did not have the opportunity to share our data with more centers. Nevertheless, compared to other studies that have reported cohorts of atypical carcinoids, our study is both one of the largest and one of the those with the longest follow-up. So we believe that our cohort size is sufficient to conclude our results.
  3. We are not aware of any other studies that have reported analyses like ours than the one we cite in the discussion. We have searched the PubMed, EMBASE, and MEDLINE. Therefore, we could only compare our results to that study.

Round 2

Reviewer 1 Report

Comments and Suggestions for Authors

Thank you for addressing refinement of the statistical analyses.